# Nutrient Regulation of Pancreatic Islet β-Cell Secretory Capacity and Insulin Production

**DOI:** 10.3390/biom12020335

**Published:** 2022-02-20

**Authors:** Kristen E. Rohli, Cierra K. Boyer, Sandra E. Blom, Samuel B. Stephens

**Affiliations:** 1Fraternal Order of Eagles Diabetes Research Center, University of Iowa, Iowa City, IA 52242, USA; kristen-rohli@uiowa.edu (K.E.R.); cierra-boyer@uiowa.edu (C.K.B.); sandra-blom@uiowa.edu (S.E.B.); 2Division of Endocrinology and Metabolism, Department of Internal Medicine, University of Iowa, Iowa City, IA 52242, USA; 3Department of Neuroscience and Pharmacology, University of Iowa, Iowa City, IA 52242, USA

**Keywords:** beta-cell function, insulin granule, proinsulin, secretory granule biogenesis, insulin secretion, granule trafficking, ER function, glutathione, Golgi

## Abstract

Pancreatic islet β-cells exhibit tremendous plasticity for secretory adaptations that coordinate insulin production and release with nutritional demands. This essential feature of the β-cell can allow for compensatory changes that increase secretory output to overcome insulin resistance early in Type 2 diabetes (T2D). Nutrient-stimulated increases in proinsulin biosynthesis may initiate this β-cell adaptive compensation; however, the molecular regulators of secretory expansion that accommodate the increased biosynthetic burden of packaging and producing additional insulin granules, such as enhanced ER and Golgi functions, remain poorly defined. As these adaptive mechanisms fail and T2D progresses, the β-cell succumbs to metabolic defects resulting in alterations to glucose metabolism and a decline in nutrient-regulated secretory functions, including impaired proinsulin processing and a deficit in mature insulin-containing secretory granules. In this review, we will discuss how the adaptative plasticity of the pancreatic islet β-cell’s secretory program allows insulin production to be carefully matched with nutrient availability and peripheral cues for insulin signaling. Furthermore, we will highlight potential defects in the secretory pathway that limit or delay insulin granule biosynthesis, which may contribute to the decline in β-cell function during the pathogenesis of T2D.

## 1. Introduction

Pancreatic islet β-cells maintain whole animal nutrient status via release of the glucoregulatory hormone, insulin, which is stored in dense-core secretory granules throughout the β-cell cytoplasm (Figure 1A). On average, a β-cell contains 10,000 insulin granules, with each granule (Figure 1B) consisting of approximately 200,000 molecules of crystallized insulin within the central core [1,2]. Insulin molecules are assembled into a hexameric arrangement stabilized by two central Zn^2+^ ions through interactions with histidine (B10) side chains (Figure 1C,D) [3,4,5]. While insulin comprises approximately 90% of the granule content, additional granule proteins are necessary to promote granule formation, maturation, and exocytosis [6,7]. These include the granin proteins, chromogranin A (CgA), chromogranin B (CgB), and VGF; the prohormone processing enzymes, prohormone convertases 1/3 and 2 (PC1/3 and PC2) and carboxypeptidase E (CPE); regulators of vesicle pH, including the vacuolar-(v)-ATPase pump and chloride channels; the zinc transporter, ZnT8; and soluble N-ethyl maleimide sensitive factor attachment protein receptor protein (SNARE) complexes containing vesicle-associated membrane protein-2 (VAMP2) and synaptotagmins 7 and 9. In sum, approximately 50–150 distinct proteins are thought to comprise the insulin secretory granule; however, little consensus exists for many of the proposed candidates due to challenges in insulin granule purification and the possible heterogeneity of insulin granule composition [8,9,10]. Upon nutrient stimulation, an isolated β-cell can secrete 5–10% of its insulin content per hour, which is triggered by coupling signals generated by oxidative metabolic cycles [11]. To sustain the secretory output of the β-cell, insulin granule biogenesis is also upregulated by metabolic cues, which affords the β-cell an immediate response to replenish insulin stores utilized during nutrient-stimulated secretion [12,13]. This coordinated regulation of insulin production by metabolic activity is a central tenant of the β-cell’s secretory program (Figure 2).

Early in response to hyperglycemia and insulin resistance, β-cells can compensate by increasing secretory capacity to boost insulin production [12,16,17]. Clinical studies highlight the ability of individuals with well-controlled blood glucose to exhibit substantial hyperinsulinemia to offset obesity and insulin resistance [18], which may involve increasing both β-cell numbers (proliferation) and secretory output (function). Increases in adult β-cell mass are relatively small, age-dependent [19,20], and the relevance to humans is less clear due to their poor proliferative capacity [21,22,23]. In contrast, human and rodent β-cells have an exceptional ability for secretory adaptations that modify insulin production in response to either nutrient deprivation or excess [12,18,24,25]. This adaptive plasticity of the β-cell’s secretory capacity may be an essential feature of long-term β-cell function and survival. While growing evidence supports a direct role for metabolic regulation of insulin expression, including *INS* gene transcription and preproinsulin mRNA translation, the mechanisms governing how metabolic signals impact ER and Golgi functions are less well-established, yet ultimately, these organelles define the β-cell’s capacity for insulin granule production [24,25,26,27,28].

Compelling evidence highlights the central role of β-cell failure (i.e., loss of glucose-regulated insulin secretion) in the transition from insulin resistance to sustained hyperglycemia and the development of Type 2 diabetes (T2D) [29,30,31]. Early signs of the failing β-cell include secretory dysfunction [29,30], such as impaired proinsulin processing, hyperproinsulinemia, and a decline in the pool of mature insulin-containing secretory granules [24,29,31,32,33,34]. In addition, metabolic defects occur, including decreased oxidative glucose metabolism and ATP production and a failure to generate metabolic coupling factors to stimulate insulin secretion [35]. Genetic predisposition (reduced secretory capacity) and environmental stresses (ER and mitochondrial stress) likely contribute to β-cell decline in T2D, yet how these events directly impact the β-cell’s secretory program are not well understood [29,30]. Potentially, the decline in metabolic function and secretory production are causally linked. As the β-cell’s metabolic program deteriorates, the generation of coupling factors wanes, which become insufficient to support the demands of the insulin biosynthetic pathway, leading to defects in ER and Golgi functions. Over time, this unresolved imbalance drives the transition from β-cell compensation to β-cell failure. In this review, we provide evidence for multiple points of metabolic inputs to regulate the insulin biosynthetic pathway, including ER and Golgi functions (Figure 2), and offer how the β-cell may use the adaptive plasticity of the secretory program in defense against nutrient challenge and chronic stress. We will also discuss possible mechanisms leading to β-cell failure when these adaptive responses become insufficient to maintain β-cell secretory function.

## 2. Nutrient-Regulated Insulin Secretion

Pancreatic β-cells promote nutrient uptake in the post-prandial state via the release of insulin, whereas, in a fasted state, β-cells suppress insulin secretion to prevent hypoglycemia. To discern fed and fasted states, the β-cell measures nutrient availability using a series of metabolic cycles that directly couple the rate of fuel oxidation with the triggering and amplification of insulin exocytosis [11]. Glucose uptake from the plasma by glucose transporters, GLUT1 and GLUT2 in human and rodent β-cells, respectively, and phosphorylation by glucokinase, serve as the first committed steps in glucose metabolism. Importantly, the activity of glucokinase defines the rate-limiting step in glucose oxidation and thereby also dictates the threshold for stimulation of insulin secretion [36,37]. Distinct from the high-affinity isoforms of hexokinase, the K_m_ of glucokinase is approximately 8 mM, which allows the β-cell to sense changes in blood glucose around the narrow physiological range of 4.4–6.1 mM glucose. Furthermore, the absence of the n-terminal domain present in other hexokinase family members, which are sensitive to product inhibition by glucose-6-phosphate, allows the β-cell’s glucokinase to respond more precisely to changes in blood glucose and provide a continuum of metabolic inputs during persistent nutrient stimulation [38].

Following glucose uptake, glucose-stimulated insulin secretion is triggered by a metabolically driven action potential [39]. Increased mitochondrial metabolism of glucose yields a rise in the intracellular ATP: adenosine diphosphate (ADP) ratio, which promotes closure of the ATP-sensitive potassium (K_ATP_) channels and leads to plasma membrane depolarization [40]. The increase in the resting plasma membrane potential from roughly −70 mV to −15 mV stimulates the opening of the voltage-gated L-type Ca^2+^-channels [39,40,41,42,43], and Ca^2+^ influx triggers the exocytotic release of plasma membrane-docked insulin granules [44] via interactions of Ca^2+^-regulated synaptotagmins and the SNARE complex [45,46]. Recent data has identified specific subpopulations of insulin granules that show compositional differences defined by the presence of either synaptotagmin 7 or 9 on the granule membrane [47]. Granules containing synaptotagmin 9, which shows high-affinity Ca^2+^ binding, exhibit fast release kinetics and may be initially utilized during first-phase insulin release (3–5 min following glucose stimulation). In contrast, granules expressing the low-affinity Ca^2+^ receptor, synaptotagmin 7, have slow-release kinetics and may be recruited during the second, prolonged phase of insulin secretion.

Although membrane depolarization and Ca^2+^ influx are necessary and sufficient to trigger β-cell exocytosis, glucose is well-described to further amplify insulin secretion beyond the direct actions of electrochemical coupling [48,49,50,51]. Additional metabolites generated via successive turns of the mitochondrial TCA cycle are thought to amplify insulin secretion in direct proportion to the nutrient stimulus and can sustain insulin release as long as the nutrient stimulus is present. While a consensus has yet to be reached on the quantitative contributions of various metabolic coupling factors [11,52], glucose-derived nicotinamide adenine dinucleotide phosphate (NADPH) and glutathione reducing equivalents (GSH) closely mirror glucose-stimulated insulin secretion and can directly stimulate exocytosis in whole-cell patch clamped β-cells [53,54,55]. Production of NADPH may stem from several anaplerotic cycles involving pyruvate [56,57] and/or glutamate [58,59,60], as well as the pentose phosphate pathway [61], which have been extensively reviewed elsewhere [11]. The reduction of glutathione by the NADPH dependent enzyme, glutathione reductase, provides a redox shuttle to activate the deSUMOylating enzyme, SUMO specific peptidase-1 (SENP1), which acts on the exocytic SNARE machinery [55], including synaptotagmin 7, syntaxin 1A and the syntaxin 1A inhibitor, tomosyn, to amplify insulin secretion [62,63,64,65]. In addition to this redox-supported pathway, mitochondria oscillate between anaplerotic and oxidative states, which is used to generate phosphoenolpyruvate (PEP) from malate via the mitochondrial isoform of phosphoenolpyruvate carboxykinase (PEPCK) [66,67,68]. Mitochondria to cytosol shuttling of PEP through an unknown mitochondrial carrier is then utilized by pyruvate kinase to generate ATP. This pathway provides continuous ATP production during nutrient stimulation, even as the β-cell reaches maximal mitochondrial oxidative phosphorylation, which is thought to provide the β-cell with a broad dynamic range to convert the glucose signal to exocytic stimulation. Notably, in addition to the role of ATP in membrane depolarization, ATP is also required to disentangle post-fusion SNARE complexes via the soluble N-ethylmaleimide sensitive factor attachment protein (SNAP) and the N-ethylmaleimide sensitive factor (NSF) complex [69] and thereby recycle plasma membrane binding sites for incoming insulin granules. Thus, generation of ATP via this pathway could serve two primary purposes in insulin exocytosis: (1) regulating membrane potential; and (2) insulin granule recruitment. As new technologies emerge, additional studies may continue to provide clarity to the quantitative contributions of these pathways in the glucose amplification of insulin secretion.

## 3. Glucose-Regulated Proinsulin Biosynthesis

Pancreatic islet β-cells exhibit a strong preference for the release of newly synthesized insulin. While the half-life of an insulin granule is estimated to be ~2.7 days [70], ~60% of secreted insulin has been synthesized within just a few hours [71]. The underlying mechanisms for this preference are not known but may involve latent remodeling of the granule membrane to remove or inactivate critical mediators of plasma membrane docking. Alternatively, the slow addition of an inhibitory moiety that allows the intracellular machinery to distinguish granules by age could also function to limit the exocytic use of aging granules. In support of these concepts, kinetic studies of insulin granules demonstrate age-dependent changes in granule mobility and cytoskeletal association, suggesting that age-dependent segregation and selection of insulin granules is an active process [72]; however, the molecular determinants of these differences are not known. Nevertheless, this point of preferential secretion of newly synthesized insulin is particularly important, in that insulin content does not necessarily correspond with the β-cell’s capacity for insulin secretion; rather, insulin content simply reflects the balance between the rates of insulin granule biosynthesis, secretion, and degradation [12,13]. For example, loss of the small GTPase, Rab3a, which is necessary for insulin granule trafficking, strongly impairs insulin secretion, yet proinsulin biosynthesis remains normal, and the levels of insulin content do not change [73]. Instead, increased insulin degradation prevents the unwanted buildup of unused insulin granules in Rab3a deficient islets. Similarly, defects in proinsulin export from the Golgi due to loss of granin proteins, CgB or VGF, have robust effects to limit insulin secretion but only modestly impact insulin content and proinsulin biosynthesis [74,75]. Thus, the preferential release of newly synthesized insulin implies that the proinsulin biosynthetic rate and insulin granule biogenesis exert a much stronger influence on the β-cell’s capacity for secretory output, than may be suggested by total insulin content.

Following increases in blood glucose, protein synthesis is increased approximately 2-fold in the β-cell, whereas proinsulin biosynthesis increases as much as 10-fold [26,76]. Initially, increased proinsulin biosynthesis was considered to occur via an autocrine feedback mechanism from insulin secretion; however, the rate of preproinsulin mRNA translation can be separated from insulin exocytosis. For example, glucose stimulation of insulin secretion occurs at 4–6 mM glucose, whereas the stimulation of proinsulin biosynthesis occurs at a lower threshold, in the range of 2–4 mM glucose [77]. Furthermore, sulfonylureas, which close the K_ATP_ channels and depolarize the β-cell’s plasma membrane, potently activate insulin release yet fail to stimulate proinsulin biosynthesis, in the absence of nutrient stimulus [78]. In contrast, diazoxide, which opens the K_ATP_ channels and prevents membrane depolarization, strongly suppresses insulin secretion in response to nutrient stimulation but does not alter proinsulin synthesis rates [79]. Collectively, these studies highlight that the coupling of insulin production with nutrient stimulation is distinct from the exocytic pathway, which may be necessary to ensure that granule replenishment exceeds granule depletion through exocytosis.

Long-term (>12 h) increases in ambient glucose levels can increase preproinsulin mRNA expression either through *INS* transcription and/or stabilizing the preproinsulin mRNA [27,80]; however, the acute response (<4 h) to upregulate proinsulin biosynthesis is direct and occurs through preproinsulin mRNA translation. A stem-loop structure in the 5′-untranslated region (UTR) of the preproinsulin mRNA was shown to be glucose-responsive and was termed the preproinsulin glucose element (ppIGE) [27]. In addition to proinsulin, nutrient regulation of at least 50 other proteins that reside within the insulin granule were documented [26]. For example, the serine endoproteases, PC1/3 and PC2, which remove C-peptide from proinsulin, and the exopeptidase, CPE, which trims the terminal arginine and lysine residues remaining in insulin after proteolytic cleavage [81], are known to be glucose-regulated and contain the ppIGE within the 5′-UTR of their respective mRNAs [13,76,82]. Similarly, the second most abundant proteins produced by the β-cell, the granin proteins, which include CgA and CgB, also contain the ppIGE within their mRNAs [13]. Though the precise metabolic coupling signals that cue the upregulation of secretory granule protein synthesis have yet to be identified, coordinating the biosynthetic regulation of key secretory granule proteins along with proinsulin would seem both logical and necessary to successfully orchestrate insulin granule biogenesis [12,13].

## 4. Proinsulin Folding in the Endoplasmic Reticulum (ER)

Proinsulin is derived from the precursor molecule, preproinsulin, following translocation from the cytosol into the ER and proteolytic cleavage of the n-terminal signal sequence via signal peptidase [83,84]. Once inside the ER, proinsulin tertiary structure is achieved by actions of ER-resident chaperones such as the immunoglobulin binding protein (BiP) and glucose regulated protein 94 (GRP94) [85,86,87] in conjunction with protein disulfide isomerases (PDIs), which catalyze the formation of the three disulfide bonds between non-sequential cysteines in the A and B chains of proinsulin at positions A7-B7, A20-B19, and A6-A11 (Figure 3) [84,88,89]. Ultimately, the balance between chaperone protein expression and availability and proinsulin biosynthetic load defines the β-cell’s ER folding capacity. The impact of disrupting this balance in the β-cell is highlighted by loss of function studies in the ER folding machinery, including the chaperone, GRP94, and the BiP co-chaperone, p58^IPK^, which result in defects to proinsulin folding and compromise insulin granule production and β-cell function [86,87,90,91]. In addition, rodent knockout models of PDIs, including Pdia1 and Prdx4, exhibit the formation of high molecular weight, disulfide-linked aggregates of proinsulin, which results in decreased insulin granule formation and glucose intolerance [88,90,92]. The consequences of proinsulin folding mutations were also demonstrated through dozens of human *INS* gene mutations that result in the syndrome known as mutant *INS*-gene-induced diabetes of youth (MIDY) [93,94]. In the majority of MIDY cases, heterozygous mutations in proinsulin folding function as dominant negative alleles. Dimerization of mutant, misfolded proinsulin with wild type proinsulin prevents the successful exit of the wild type proinsulin from the ER [95]. The subsequent buildup of proinsulin aggregates becomes proteotoxic to ER function and ultimately leads to β-cell loss from apoptosis [96]. In healthy β-cells, up to 20% of nascent proinsulin may be misfolded, whereas in the progression of T2D, the formation of misfolded proinsulin aggregates increases [97]. Whether this increase in misfolded proinsulin in T2D β-cells is pathological, akin to MIDY, and contributes to the demise of the β-cell, or whether it is simply a consequence of increased proinsulin biosynthesis from chronic hyper-nutrient stimulation, remains to be determined.

## 5. Nutrient Sensing of Client Protein Load in the ER

ER folding capacity is regulated, in part, via three ER transmembrane receptors, protein kinase R-like ER kinase (PERK), inositol requiring enzyme 1α (IRE1α), and activating transcription factor (ATF6α), which oversee the ER stress response program (also referred to as the unfolded protein response, UPR) [99,100,101]. Activation of PERK can both attenuate global protein synthesis rates via phosphorylation of eIF2α on serine 51, which pauses the influx of client proteins into the ER, and activate a transcriptional program through translational control of ATF4 and the subsequent upregulation of C/EBP homologous protein (CHOP) [101,102,103,104,105,106]. The actions of IRE1α primarily occur through the unconventional, cytoplasmic splicing of X-box binding protein-1 (XBP-1) mRNA to remove an intronic segment resulting in a frameshift, which allows the synthesis of the active XBP-1 (s) transcription factor [107,108,109]. Last, trafficking of ATF6α from the ER to the Golgi results in site-1 and site-2 protease (S1P and S2P) cleavage to release the cytosolic fragment, which functions as a basic leucine zipper (bZIP) transcription factor [110]. Collectively, the primary function of the ER stress response program is to sense client protein overload through the loss of available chaperone protein binding (i.e., accumulation of misfolded proteins) and adjust ER folding capacity via the activation of transcriptional and translational programs that modulate the expression of ER chaperones, PDIs, and glycosylation enzymes [100,106,111]. A detailed discussion of the contribution of these sensors to β-cell function and their transcriptional programs is beyond the scope of this review and was expertly reviewed elsewhere [99,112].

As described earlier, β-cells increase proinsulin biosynthesis in response to elevated glucose metabolism, which may account for as much as 30–50% of total protein synthesis in the β-cell [26,97]. Pseudo-time ordering of single-cell RNAseq data demonstrates that healthy β-cells oscillate between states of high proinsulin synthesis and the subsequent activation of the ER stress response, which may function as a critical feedback mechanism to adapt the β-cell’s ER folding capacity to meet the demands of nutrient stimulated increases in proinsulin biosynthesis [28]. This essential feature of β-cell plasticity allows for compensatory changes to boost secretory output and overcome insulin resistance early in T2D via the expansion of the secretory pathway [12,18,113]. In support of this, the IRE1α/XBP-1 (s) transcriptional program is required for β-cells to increase insulin production as a compensatory response to nutrient overload in rodent dietary models [114]. Similar requirements for matching ER folding capacity with secretory output were also demonstrated for other professional secretory cells, such as B-cell to plasma cell differentiation [115,116,117] and in the development of pituitary cells [118]. As T2D progresses and the demand for insulin increases, the limitations of this feedback loop become apparent. Ultrastructural analysis of β-cells from human T2D and rodent models of diabetes demonstrate a profound expansion of the rough ER membranes, supporting the idea that T2D β-cells attempt to fulfill the secretory demand via increasing ER size and folding capacity [12,24,32,119]. Furthermore, decreasing proinsulin synthesis via the deletion of one or both of the insulin genes in mice (*INS1* and *INS2*; human insulin is encoded by a single gene, *INS*) was shown to greatly attenuate markers of the ER stress response in models of nutrient overload [120]. Thus, the balance between client protein (proinsulin) load and ER folding capacity is a critical feature of the β-cell’s secretory program to adjust insulin production with the demands for insulin release; however, repeated and/or long-term stress may eventually overcome the β-cells adaptive capacity leading to irreversible damage and β-cell demise [121].

## 6. Metabolic Influence on β-Cell ER Redox Homeostasis

Disulfide bonds are essential to maintain the folded structure of secretory and membrane proteins. In the highly oxidizing environment of the ER lumen, disulfide bonds readily form [88]; however, in some instances, the mispairing of cysteines occurs and requires isomerization by ER resident PDIs to correct. Disulfides arising from non-sequential cysteine residues, such as the three disulfide bonds present in proinsulin (Figure 3), are particularly prone to mispairing and rely on PDIs to facilitate proper proinsulin folding [90,122]. To reset PDIs for sequential rounds of client protein isomerization, ER oxidoreductin 1α (Ero1α) utilizes a highly regulated feedback loop of intramolecular disulfide switches involving glutathione disulfide (GSSG) and reduced glutathione (GSH) redox shuttles [123,124,125]. In addition to glutathione, thioredoxin also acts as an electron donor in the isomerization of disulfide bonds that reset PDIs [126], although the mechanisms regulating the transfer of reducing equivalents from the cytosol to the ER lumen is not well understood [126,127]. While the quantitative contribution of these two donor systems (glutathione and thioredoxin) to disulfide bond formation has yet to be determined, both mechanisms are regulated by NADPH-dependent enzymes, namely glutathione reductase and thioredoxin reductase, which are sensitive to fluctuations in mitochondrial metabolism [126]. In hepatocytes, TCA metabolic flux directly alters glutathione redox and impacts ER oxidation status [128]. Increased metabolic activity facilitates GSH reduction by glutathione reductase (Gsr), resulting in a more reduced ER lumen, whereas decreasing TCA activity has the opposite effect—diminishing GSH availability and oxidizing the ER environment. In β-cells, NADPH and glutathione redox fluctuate with physiological changes in glucose metabolism [53,54,55]. Thus, in addition to metabolic regulation of insulin secretion, the oxidative protein folding capacity of the β-cell’s ER may also be metabolically responsive and rely on reducing equivlaents to support increased proinsulin biosynthesis (Figure 4). Indeed, the loss of the primary NADPH-producing enzyme in β-cells, cytosolic isocitrate dehydrogenase 1 (Idh1), not only decreases available GSH and impairs insulin secretion [11,55,57] but also diminishes insulin granule formation [129]. While it remains to be determined if Idh1 loss results in a direct effect on ER oxidative protein folding capacity, proinsulin trafficking and insulin granule formation can be restored by the addition of cellular reducing equivalents. Given that the synthesis of approximately 50 insulin granule proteins is glucose-regulated [26], priming the ER oxidative protein folding machinery to accept an increase in protein client load via metabolically supplied redox equivalents may be an elegant mechanism to ensure the efficient maturation of secretory proteins that support insulin granule biogenesis.

Based on this model of metabolic regulation of ER redox homeostasis (Figure 4), a disturbance in the available electron flow to reset PDIs, such as a defect in mitochondrial metabolism, coupled with an increase in protein client demand would lead to a pathological decrease in the folding capacity of the ER and subsequent impairments in insulin granule synthesis [130]. Indeed, the formation of aberrant intermolecular disulfide bonds between proinsulin molecules was documented in T2D β-cells, suggesting that impaired disulfide bond isomerization in the ER contributes to the insulin deficiency in the progression of T2D [85,92,97,122]. Mitochondrial dysfunction resulting in alterations to glucose metabolism, including decreased NADPH and GSH production, are hallmarks of T2D β-cells [29,35,55], supporting the idea that the availability of redox equivalents may be limited by impaired metabolic capacity and/or increased usage for neutralizing reactive oxygen species. In contrast, antioxidant treatment using butylated hydroxyanisole or supplementation with reducing agents can improve proinsulin folding and promote insulin granule formation in diabetes models [91,114,129]. Thus, while the hyperglycemia observed in T2D β-cells may signal increased proinsulin synthesis, metabolic dysfunction fails to provide the necessary redox shuttles to support ER functions. Collectively, these events lead to ER redox imbalances that greatly hinder proinsulin disulfide bond formation and folding and thereby contribute to defects in insulin granule formation.

Metabolic alterations were also reported early in the pathogenesis of Type 1 Diabetes (T1D) [131]. Following exposure to proinflammatory cytokines, disruptions to mitochondrial metabolism, such as decreased oxygen consumption, generation of free radicals, and oxidative stress, are associated with substantial fragmentation of mitochondrial networks [132,133,134]. Evidence of secretory dysfunction in T1D etiology in the form of neoantigens, identified as hybrid peptides containing fragments of insulin, C-peptide, IAPP, and/or CgA, suggest that alterations to prohormone trafficking and processing contribute to the development of CD4^+^ autoimmunity early in disease progression [135,136,137]. Non-native disulfide bond formation during proinsulin folding in the ER was suggested as a contributing factor to neoantigen formation [138]. While a definitive link between cytokine exposure, non-native disulfide bond formation, and defects in proinsulin trafficking have yet to be demonstrated in T1D pathogenesis, future studies may consider this common theme of mitochondrial dysfunction and ER redox status as a possible mechanism contributing to T1D etiology.

## 7. Proinsulin Sorting in the Golgi

Following oxidative protein folding in the ER, proinsulin is trafficked into the Golgi, where it is packaged with other regulated secretory cargo, such as the processing hormones, PC1/3, PC2, and CPE, into the budding secretory granule [7,83]. Although the process of protein cargo selection and sorting into the budding granule is not well understood, aggregation of the granule cargo in the late Golgi and/or *trans*-Golgi network (TGN) is considered a rate-limiting step in the sorting process [139,140]. Indeed, proinsulin is known to accumulate in the *trans*-Golgi cisternae, presumably being delivered into budding granules [141]. Active sorting of granule cargo via sorting receptors was proposed to occur in the TGN through a process termed sorting by entry [142,143]; however, support for specific sorting receptors has waned, in part, due to the lack of conserved topological sorting signals within proinsulin and other granule proteins, which is a common problem faced by most regulated secretory proteins [140]. As an alternative, selective protein aggregation was suggested to promote proinsulin and other regulated cargo sorting independent of a specific sorting receptor. In this model, biophysical determinants in the protein structure rather than sequence elements dictate self-assembly and sequestration. Granin proteins, including CgA, CgB, and VGF, which are major constituents of secretory granules in multiple endocrine tissues and are known to self-aggregate, were suggested to promote granule biogenesis in the TGN [139,144,145,146,147]. Through aggregation, granin proteins selectively capture secretory cargo for delivery to the budding granule [147,148]. Despite this, the direct association between granin proteins and secretory cargo is challenging to detect, in part, because these interactions are likely to be low affinity and transient, occurring only in the late Golgi. Recently, the aggregation model of protein sorting was challenged, in part, because the in vitro conditions used to demonstrate granin protein aggregation of millimolar Ca^2+^ and pH 5.5 are more extreme than those present in the TGN lumen, which is likely to be approximately 100 μM Ca^2+^ and pH 6.1–6.3 [140,149]. In addition, the formation of solid aggregates containing proinsulin and other secretory granule cargo would likely preclude endoproteolytic access of dibasic cleavage residues necessary for proinsulin to insulin conversion within the maturing granule, yet insulin conversion is highly efficient [1,71,83]. An updated mechanism suggests that granin proteins facilitate protein sorting in the TGN through liquid phase separation [149]. Notably, liquid phase separation occurs in intrinsically disordered proteins with low sequence complexity, such as granins, whereas solid aggregates are more likely to form through β-sheet interactions and/or repetitive sequences, which are absent in granin protein structure [149,150]. Through multivalent interactions, intrinsically disordered proteins can self-assemble into supramolecular complexes that function as scaffolds to promote macromolecular crowding and/or the segregation of neighboring proteins [150,151]. The fluidity of these complexes is highly influenced by the surrounding environment (pH, ions) and does allow for protein exchange. While the challenge of protein sorting fidelity in the TGN lumen still remains, the model of liquid phase separation for protein segregation is consistent with the notion that granule condensates remain accessible to processing enzymes for the conversion of proinsulin to insulin during granule maturation [1,81,83].

The granin proteins, CgB and VGF, have emerged as key determinants of granule biogenesis in the β-cell [74,75,152]. β-cell-specific deletion of VGF results in defective insulin release due to a lag in granule replenishment accompanied by an accumulation of granule proteins in the TGN, including proinsulin, CgA, and CgB [74]. In addition, loss of CgB results in a significant trafficking delay in proinsulin Golgi export and the delayed appearance of newly synthesized insulin granules at the plasma membrane [75], leading to reduced insulin secretion and proinsulinemia [75,152]. Multiple studies in various cell types, including β-cells, PC12 pheochromocytomas, and adrenal chromaffin cells, showed that VGF directly interacts with CgB [75,153]. Interactions between proinsulin, VGF, and CgB were also recently demonstrated through ascorbate peroxidase 2 (APEX2) proximity labeling [129]. Based on the recently proposed model of liquid phase separation for protein segregation in the TGN, highly disordered regions of CgB may function as a “cargo sponge” through multivalent interactions with client proteins, such as proinsulin, and promote phase separation from other soluble proteins not destined for the insulin secretory granule [149,151]. Whether VGF has a similar ability was not tested, but VGF does contain long segments of intrinsically disordered regions lacking any recognizable secondary structure (Figure 5). Taken together, these data posit VGF and CgB as critical regulators of granule protein exit from the Golgi in the islet β-cell.

## 8. Golgi Assembly of the Insulin Granule

Cholesterol and sphingolipid rich microdomains in the TGN membrane were proposed as critical sites for concentrating secretory granule cargo into condensates to promote granule budding [156,157,158,159,160,161]. Chemical depletion of cholesterol and/or the genetic inactivation of Golgi cholesterol transporters, oxysterol binding protein (OSBP), ATP binding cassette subfamily G1 (ABCG1) and ATP binding cassette subfamily A1 (ABCA1), negatively impact multiple steps in insulin granule biogenesis, including granule budding, proinsulin processing, and granule stability [160,162,163,164]. Secretogranin 3 (Sg3), which is required for insulin granule formation, associates with the granule membrane via direct interaction with cholesterol and serves as an adaptor to anchor other granule proteins, such as CgA and Sg2 [165,166]. CgB, CPE, PC1/3, and PC2 were also shown to associate with detergent-resistant lipid rafts in the granule membrane, though whether the interaction is direct or via an adaptor remains unknown [167,168,169,170,171].

In addition to serving as membrane binding sites, lipid microdomains may also have a more active role in granulogenesis [140,159]. Sphingomyelin is known to activate the TGN-resident Ca^+2^ ATPase, secretory pathway Ca^2+^-ATPase pump type 1 (SPCA1), which regulates Ca^2+^ entry in the Golgi lumen [172]. Deficiency of SPCA1 impairs Golgi functions, such as cargo trafficking, reduces insulin secretion due to alterations in insulin granule maturation, and reduced expression of SPCA1 occurs in models of T2D [172,173]. SPCA1-regulated Ca^2+^ entry is proposed to activate oligomerization of the Golgi resident Ca^2+^ binding protein 45 (Cab45), which promotes the sorting of regulated secretory cargo [172,174,175,176]; however, the contribution of Cab45 to insulin granule biogenesis has yet to be reported. Potentially, higher-order assembly of granins at TGN budding sites (Figure 6), which are rich in cholesterol, facilitate phase separation and concentration of granule cargo into the budding vesicle [156]. Sphingomyelin activation of SPCA1 may lead to a local increase in Ca^2+^ in step with glucose-dependent oscillations in Ca^2+^ influx, which enhance liquid phase separation of granin-bound complexes containing secretory cargo [140,159]. In addition, Ca^2+^-activated oligomerization of Cab45 may further drive the condensation of granule cargo and prompt vesicle budding from TGN exit sites [140,172,174]. 

## 9. Nutrient Regulation of Golgi Function

As discussed earlier, increased nutrient-stimulated proinsulin biosynthesis can overcome the ER’s capacity to efficiently fold proinsulin and lead to the activation of an ER stress response program [16,28,97,99,121,122]. Oscillations between proinsulin biosynthesis and ER stress were documented in human β-cells and are likely to be a natural adaptation to changes in nutrient status [28]. Similar to the ER, Golgi size and function can also be regulated by nutrient availability [12,25]. Increased Golgi size was documented in β-cells from human T2D and genetic rodent models of diabetes [12,24,32]. Whether a parallel program exists to prompt Golgi expansion in response to nutrient challenge and/or increased protein client load is less clear, in part, because the basic Golgi stress response is not well understood in mammalian cells and may partially overlap with the ER stress response program [177]. Recently, a Golgi stress response signature was identified in T1D and T2D islets using a bioinformatics approach by comparison to the stress response elicited by the pharmacological agent, brefeldin A, which drives the collapse of the Golgi structure into the ER. ATF3, ADP ribosylation factor-4 (ARF4), cAMP response element binding protein-3 (CREB3), and Component Of Oligomeric Golgi Complex-6 (COG6) were identified as potential Golgi regulatory factors that are dysregulated in both major forms of diabetes [178]. Of particular interest is the CREB3 family of stress response bZIP transcription factors. Similar to ATF6, CREB3 family members are expressed as inactive transmembrane proteins localized to the ER membrane and upon stimuli, such as Golgi stress, traffic to the Golgi [179], where the cytosolic portion is proteolytically cleaved by S1P and S2P to release the active transcription factor [180]. CREB3 proteins are required to increase secretory output in the development of professional secretory cells [181,182], such as antibody-secreting plasma cells [183] and pituitary cells [118], through the regulation of Golgi homeostasis [179,182]. In human β-cells, CREB3 is upregulated by palmitate and attenuates palmitate toxicity [184], potentially acting as a mediator of the β-cell Golgi stress response [178,185] independent of the classic ER stress response genes [184,186,187]. Although the precise criteria for CREB3 activation remain to be determined, the Golgi stress response via CREB3 may be analogous to the ER stress response and elicit a coordinated program that expands Golgi size and sorting capacity. This response could be dictated by client protein overload in the Golgi directly, akin to the UPR, which overcomes the Golgi’s ability to efficiently sort and export granule proteins; however, given that CREB3 is an ER-localized protein, it is unclear how Golgi signals would relay back to the ER. Alternatively, Golgi stress response activation may be a preemptive mechanism elicited from the ER that recognizes the need to simultaneously elevate the functional capacity of both major secretory compartments to accommodate nutrient-activated increased proinsulin synthesis.

Improving peripheral insulin sensitivity and restoring glucose homeostasis through either pharmacological intervention or caloric restriction can remove the persistent hyperglycemic stimulus exerted on the β-cell secretory program [12,16,17]. In response, the β-cell normalizes nutrient-stimulated biphasic insulin release and repopulates the insulin secretory granule pool [24,188]. Using a phosphoproteomics screen, the Golgi resident kinase, Family With Sequence Similarity 20 Member C (FAM20C), surfaced as a potential mediator of this β-cell transition between states of nutrient overload and recovery in which decreased FAM20C activity was associated with β-cell recovery [189]. Although FAM20C is considered to have a broad range of targets [190] and is potentially able to phosphorylate a majority of the β-cell secretome, Cab45 was recently identified as a FAM20C substrate [191]. The phosphorylation of Cab45 on five distinct serine/threonine residues destabilizes Cab45 oligomers independent of Ca^2+^ binding and promotes cargo exit from the TGN. In contrast, the inactivation of FAM20C kinase activity or removal of the Cab45 phosphosites negatively impacts the Golgi exit of regulated secretory proteins. While the contribution of FAM20C regulation of Cab45 to β-cell insulin granule biogenesis has yet to be established, nor is it known how nutrient status regulates FAM20C activity, FAM20C could serve a critical role in defining the rate of secretory cargo exit from the Golgi (Figure 6). Through cycles of Cab45 oligomerization and cargo sequestration followed by phosphorylation and oligomer disassembly, FAM20C activity may contribute to a quality control mechanism to ensure capture of the appropriate granule cargo into the budding insulin secretory granule [191]. In addition, nutrient regulation of FAM20C activity may be used to match insulin granule biogenesis with nutrient demands [189]. Decreased FAM20C activity would delay granule cargo exit from the Golgi, which may occur in nutrient depleted or fasted states, in which the demand for insulin granule production is low. In contrast, increased FAM20C activity in response to nutrient stimulation would expedite granule cargo exit to replenish insulin granule stores during increased demand for insulin granule exocytosis.

## 10. T2D Therapies: What Can We Learn about β-Cell Plasticity?

Multiple modalities exist for treating hyperglycemia and/or insulin resistance in T2D, which can either increase insulin release or relieve the secretory burden of the β-cell. Relative to their modes of action, these agents have distinct impacts on β-cell health and survival, which have proven useful in understanding the limits of β-cell function and adaptive compensation [12,16,17]. Early diabetes therapies, such as the sulfonylureas, targeted hyperglycemia by directly stimulating insulin release independent of glucose. While effective at elevating plasma insulin and lowering glycated hemoglobin A1c [192], sulfonylureas are not protective to the β-cell and, in fact, may accelerate β-cell demise and progression to overt diabetes [193,194,195]. Incretin-based β-cell therapies, such as the glucagon-like peptide-1 receptor (GLP-1R) agonists, directly potentiate insulin secretion via adenosine 3^′^, 5^′^-cyclic monophosphate (cAMP)-mediated pathways [196,197] as well as enhance β-cell glucose metabolism [198], but unlike sulfonylureas, only stimulate insulin release in response to nutrient stimulation [197]. GLP-1R agonists can also attenuate β-cell loss both through β-cell-specific effects, such as reduced ER and oxidative stress, as well as acting peripherally to decrease the expression of proinflammatory cytokines and promote weight loss through appetite suppression [199,200]. The long-term durability of GLP-1R agonists in maintaining β-cell function remains to be determined, but recent data suggest that these agents may also lead to β-cell burnout [201]. In contrast to β-cell-directed therapies, insulin-sensitizing agents, such as thiazolidinediones (TZD) and, to a lesser extent, metformin, can delay the loss of islet β-cell mass in rodent diabetes models and promote recovery of β-cell function in human T2D, despite primarily functioning to improve insulin signaling in peripheral tissues and thus indirectly impacting β-cell function [195,202,203]. Similarly, normalizing blood glucose with the sodium/glucose cotransporter 2 inhibitor, empagliflozin, which promotes urinary excretion of glucose to attenuate hyperglycemia, was shown to directly promote β-cell recovery, including repopulation of the mature insulin secretory granule pool [188]. More direct studies documented that reducing β-cell output via insulin (glargine) supplementation or acute diazoxide treatment can improve β-cell responses and glycemic control [204,205,206]; however, concerns over hypoglycemic episodes and weight gain preclude the recommended use of basal insulin therapies as a single agent treatment for T2D [207]. Recently, polyagonist peptides have emerged as potential T2D therapies, which use carefully designed unimolecular peptides, rather than multivalent, bi- or trifunctional ligands, to independently activate the two incretin receptors, GLP-1R and the glucose-dependent insulinotropic receptor (GIPR), and the glucagon receptor (GcgR) [208,209]. Thus far, these dual and triagonist peptide agents show superior benefits to promote weight loss and glycemic control in clinical studies compared to single peptide-based therapies and thus may be able to promote β-cell health as well as relieve the peripheral insulin demand [210,211,212,213]; however, the long-term impact of these polyagonist peptides on β-cell function is not yet known. The observation that weight loss alone can lead to a marked improvement in β-cell function has spurred continued interest in anti-obesity therapies as a key goal in developing more effective T2D therapies [214,215]. Collectively, these observations suggest that relieving the demand for insulin release via independent mechanisms can promote β-cell health and improve long-term outcomes [12,17], whereas β-cell exhaustion from persistently elevated insulin secretion may be an underlying cause of dysregulated secretion in T2D and contribute to the final transition to β-cell failure [17,29,216]. These unresolved issues represent a significant gap in our understanding of β-cell decline in T2D and demonstrate the urgent need to continue investigations into pathways that preserve glucose-controlled β-cell functions as therapeutic approaches.

## 11. Concluding Remarks

In summary, we have discussed how metabolic fuel sensing functions at multiple levels in the pancreatic islet β-cell to coordinate insulin secretory capacity with the physiological demands of nutrient status [12]. ATP and other metabolic signals generated by glucose oxidation directly trigger and amplify insulin granule exocytosis through membrane depolarization and insulin granule recruitment [48,49,50]. In addition, the mitochondrial metabolism of glucose and other fuels can directly enhance preproinsulin mRNA translation [13,26,27], while metabolism-derived redox shuttles support disulfide bond formation and isomerization to ensure successful folding of nascent proinsulin polypeptides [122,129]. Similarly, Golgi export functions may be dependent on nutrient signals that can either accelerate or pause the assembly of secretory granule cargo into the budding granule to favor an excess of insulin granule production relative to insulin release [189,191]. In addition to these direct effects coupling nutrient metabolism with secretory function, sustained changes in nutrient status can also elicit global changes in the β-cell’s secretory program. Prolonged fasting can attenuate the β-cell’s secretory capacity by limiting insulin production and increasing insulin turnover to prevent hypoglycemia [25], whereas the expansion of ER and Golgi membranes seems to be a common adaptation to insulin resistance and episodic hyperglycemia [24,28,33,119]. Tentatively, the mechanisms governing how metabolic signals are integrated within the secretory program may define the limits of β-cell compensation and thereby dictate the onset of pathological states leading to β-cell failure. These mechanisms may involve direct metabolic support via intermediates such as redox coupling factors to enhance organelle functions, metabolic cues that promote organelle expansion, and/or organelle specific sensors that measure changes in protein client load and activate feedback loops to adjust organelle capacity and function. Thus, continued investigations into how nutrient signals are used by the β-cell to modulate and support secretory functions are of critical significance to our basic understanding of the β-cell’s long-term strategy for the maintenance of glucose homeostasis as well as to provide clinically relevant pathways that could be explored for diabetes therapies.

## Figures and Tables

**Figure 1 biomolecules-12-00335-f001:**
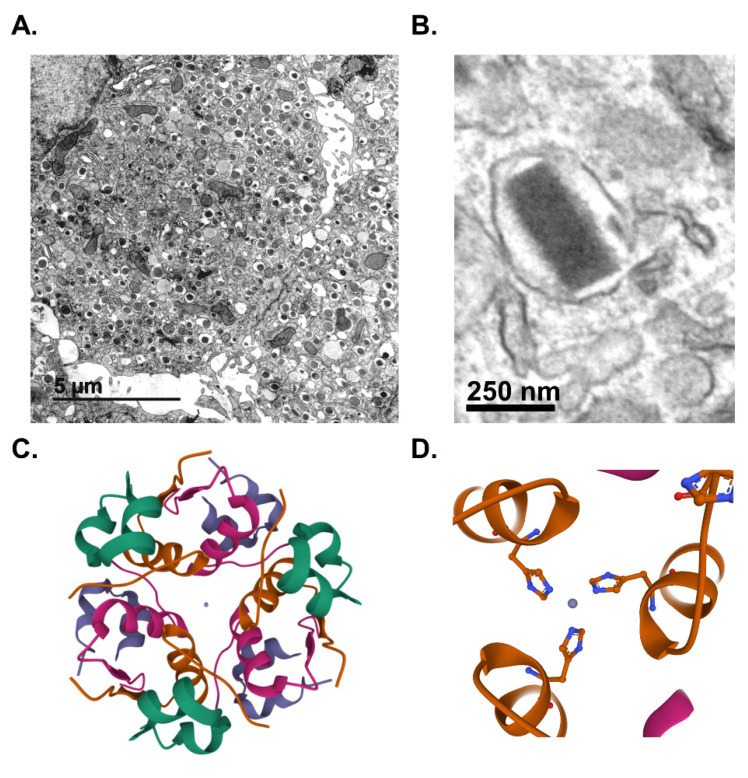
Insulin is stored in dense-core secretory granules. Transmission electron micrographs of mouse β-cells depicting dense-core secretory granules (**A**) or a single insulin secretory granule from a human β-cell (**B**). Crystal structure of hexameric insulin with central Zn^2+^ ions in gray (**C**) coordinated by histidine (A10) side chains (**D**). (**C**,**D**) Paired insulin dimers within the hexamer are colored separately for visual clarity. Lavender or green denote the A chains; purple or green denote the B chains. Images were created using Mol* of 2INS from the Research Collaboratory for Structural Bioinformatics Protein Data Bank (RCSB PDB) [14,15].

**Figure 2 biomolecules-12-00335-f002:**
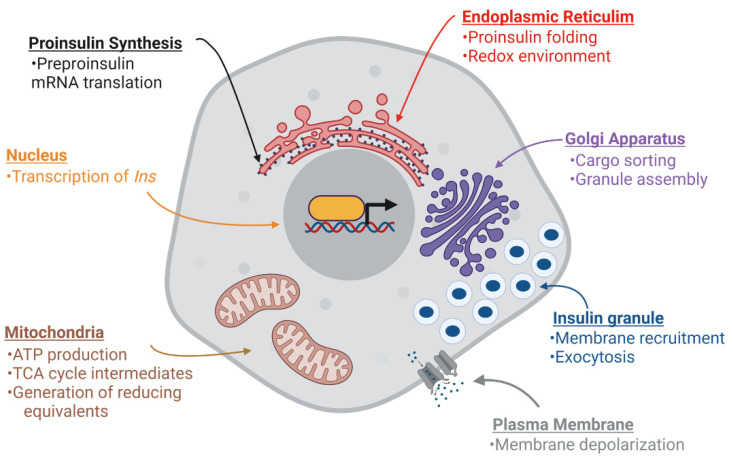
Nutrient metabolism supports β-cell function. Multiple stages of insulin production and secretion are regulated by mitochondrial metabolism through the generation of signaling molecules, adenosine triphosphate (ATP), tricarboxylic acid (TCA) cycle intermediates, and reducing equivalents. Long-term (>12 h) glucose stimulation can enhance *INS* expression through transcription and preproinsulin mRNA stability. Acute (<4 h) glucose stimulation can activate preproinsulin mRNA translation as well as support proinsulin folding in the ER by supplying reducing equivalents, such as glutathione. In addition, glucose can facilitate insulin granule assembly and export from the Golgi. Finally, glucose-derived signals can trigger plasma membrane depolarization and promote insulin granule fusion and insulin exocytosis. Images were created using Biorender.

**Figure 3 biomolecules-12-00335-f003:**
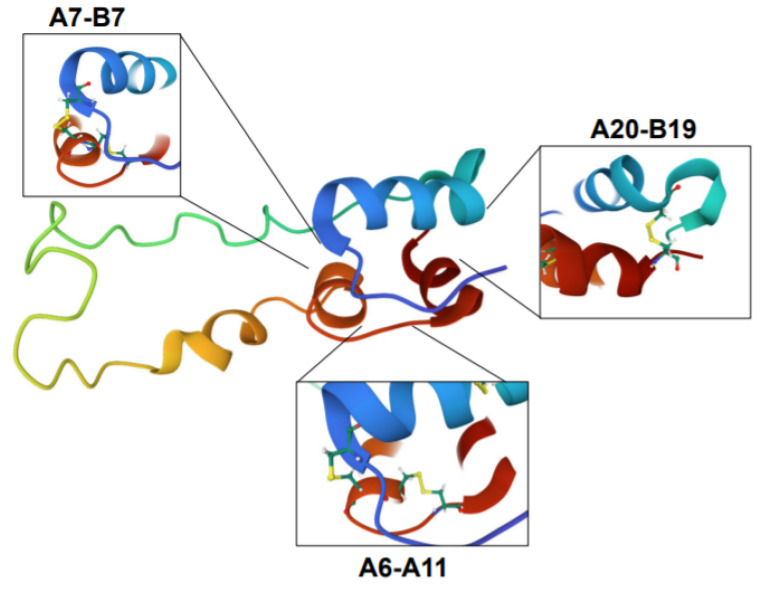
Proinsulin structure. Structure of proinsulin highlighting the disulfide bonds between the A and B chains (A7-B7, A20-B19) and within the A chain (A6-A11), which are necessary for proper folding. Structure is colorized to highlight the following features: blue is the B chain; red is the A chain; green, yellow, and orange denotes C-peptide region. Images were created using Mol* of 2KQP from the RCSB PDB [15,98].

**Figure 4 biomolecules-12-00335-f004:**
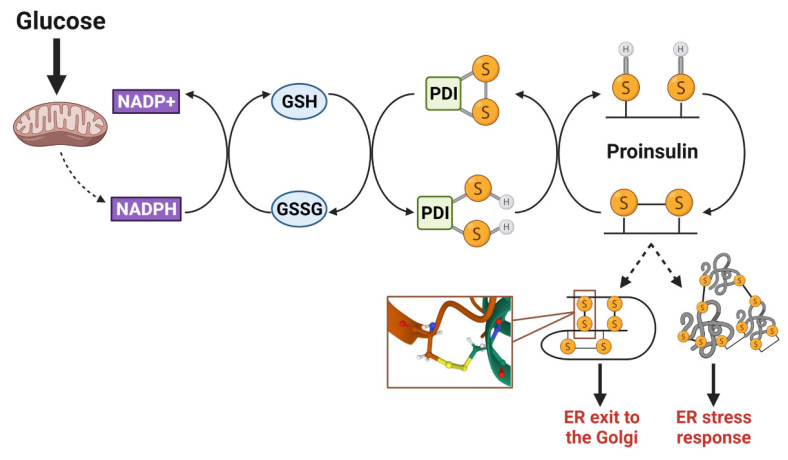
Proinsulin disulfide bond formation requires a redox shuttle. Isomerization of proinsulin disulfide bonds occurs through a metabolism-driven redox shuttle, involving NADPH/NADP+, glutathione (GSH/GSSG), and protein disulfide isomerases (PDIs). This process is necessary to correct mispaired cysteine residues that may lead to the formation of intermolecular disulfide bonds between proinsulin molecules. Upon proper folding and structural stabilization via the correct disulfide bonds, proinsulin exits from the ER and continues through the secretory pathway. SH denotes a free sulfhydryl in cysteine side chain; S-S denotes oxidized cystines in the disulfide bond. Images were created using Biorender.

**Figure 5 biomolecules-12-00335-f005:**
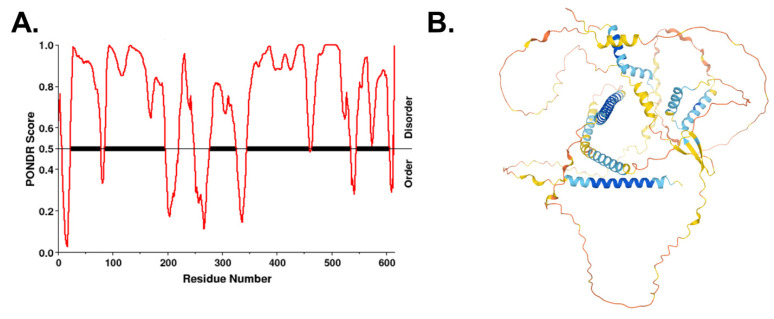
VGF is an intrinsically disordered protein. (**A**) Disordered regions of human VGF were identified using the VL-XT algorithm in PONDR. (**B**) Predicted folded structure of VGF was generated using AlphaFold [154,155]. Disordered regions lack secondary structure.

**Figure 6 biomolecules-12-00335-f006:**
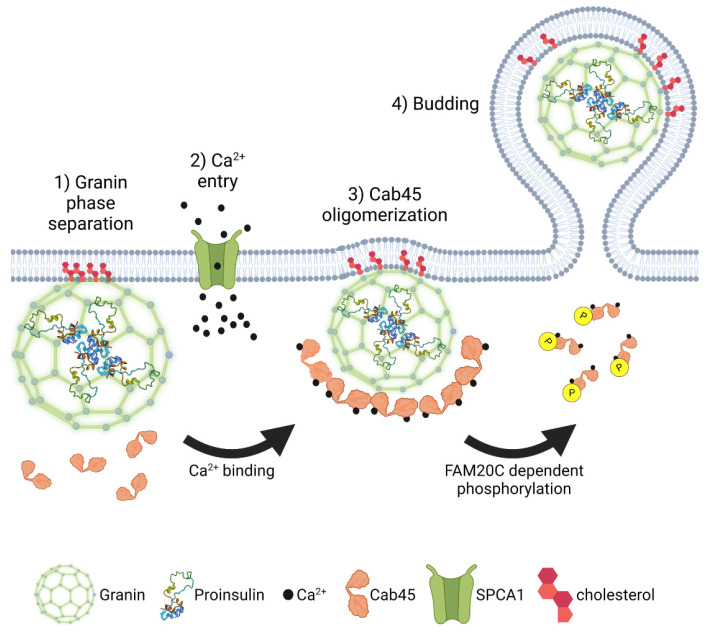
Proinsulin granule export from the TGN. Granin proteins self-assemble into multimeric macromolecular structures, which selectively capture proinsulin and other secretory granule cargo through a process referred to as liquid phase separation. As the condensate coalesces, granin proteins interact with the Golgi membrane at cholesterol-rich lipid rafts. SPCA1-mediated Ca^2+^ influx drives oligomerization of Cab45 and further condensation of granin-proinsulin complexes. Following nutrient stimulation or other signals, disruption of Cab45 oligomers through FAM20C-mediated phosphorylation promotes proinsulin vesicle budding and exit from the TGN. Images were created using Biorender.

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
