# Peer review of "Nutrient Regulation of Pancreatic Islet β-Cell Secretory Capacity and Insulin Production"

_biomolecules, 2022, doi:10.3390/biom12020335_

Round 1

Reviewer 1 Report

This manuscript is well written and describes the molecular mechanisms underlying insulin biosynthesis and secretion very thoroughly. I only have a few  comments.

  1. The title is somewhat misleading. Although nutrient regulation is discussed throughout the article, I do not feel this is the actual topic. It is rather a comprehensive review on regulation of insulin cellular production and release independent of which regulatory mechanism. I suggest to rethink the title somewhat.
  2. Row 357-369 speculates that proinsulin folding may have a causal role in T1D. The rest of the paper mainly focuses on T2D and I do not think this speculative short section belongs here. The underlying data comes from the NOD mouse, a notoriously untranslatable animal model. Unless some translatable human data exist and can be referenced, I think this section should be removed.
  3. The Concluding remarks section is surprising. It almost feels like a mini review of T2D therapies. I would haver expected an analyses of what all the mechanistic detail that has been given means for the field? Where can we take all this knowledge? Where are the putative novel drug targets in this pile of information? I do not see that the current section gives any suggestions or recommendations for important further advancements. It needs to be rewritten. Parts of the section on current therapies would fit better as an introduction to lead the reader into the review and explain why it is important to read this review.

Author Response

1) We have amended the title to better reflect the content of the review

2) We have removed the mention of the NOD mouse and provided additional comments related to human T1D

3) We have amended the section on T2D therapies to better reflect how this information leads to a better understanding of beta-cell adaptations. In addition, this section was renamed, and a new section was written for the concluding remarks to provide perspective.

Reviewer 2 Report

This manuscript is a review article, aimed to analyse current literature in remarkably interesting and important theme, such as function and plasticity of pancreatic islet β-cells. The authors emphasised the features and capacity of these cells to adapt to various nutrients and stimuli, as well as highlight potential defects in the secretory pathway that limit or delay insulin granule biosynthesis, which may contribute to the decline in β-cell function during the pathogenesis of T2D. In that sense, the article has clinical importance for pathogenesis of type 2 diabetes as well as type 1 diabetes, and possibilities of modulation and summarise broad spectrum of up to date data. In order to clarify this issue, the authors have done the complex work of critical review of sufficient number of published studies. In that context, they analysed data clearly and thoroughly. Figures are clear and well described. I suggest to the authors to add more data about the effect of GLP 1RA on remodelling of pancreatic islet β cell secretory function.

Author Response

As suggested, we have provided additional information describing the impact of GLP-1R agonists as well as the dual and triagonists as possible breakthrough T2D therapies.